# Body Size Awareness and Modular Self-Representation in Reedfish (*Erpetoichthys calabaricus*): Near-Field Passability Judgments

**DOI:** 10.3390/ani15223231

**Published:** 2025-11-07

**Authors:** Ivan A. Khvatov

**Affiliations:** Center for Biopsychological Studies, Moscow Institute of Psychoanalysis, 121170 Moscow, Russia; ittkrot1@gmail.com; Tel.: +7-926-339-23-00

**Keywords:** body size awareness, self-representation, modular cognition, body-as-obstacle, reedfish (*Erpetoichthys calabaricus*), fish cognition, lateral line, near-field sensing, affordance-based decision making, benthic ecology

## Abstract

**Simple Summary:**

Body size awareness is an animal’s ability to take its own dimensions into account when negotiating obstacles. We examined this ability in reedfish, a long, bottom-dwelling species. In a tank we installed a partition with three openings. In Experiment 1 all three openings were passable but differed in size. In Experiment 2 only one opening was passable, whereas two larger ones were not. Fish first approached any opening, but they attempted to pass almost exclusively through the truly passable one; after such attempts, they always entered the target compartment. When all openings were passable, choices were at chance: fish did not favor the largest opening but simply used one that was “big enough”. These findings indicate that the pass/not-pass decision is made at close range, likely using tactile and hydrodynamic cues from the head. The study extends evidence for body size awareness to a phylogenetically distant fish and may inform better husbandry and environmental enrichment for aquatic species, as well as bio-inspired navigation strategies in confined spaces.

**Abstract:**

Body size awareness—a component of bodily self-representation—allows animals to match their own dimensions to environmental constraints. This study tested whether reedfish (*Erpetoichthys calabaricus*), a benthic ray-finned species with limited vision, can evaluate aperture passability relative to their body size. Eight fish performed a “body-as-obstacle” task. After training, each individual completed 36 trials in Experiment 1 (three passable circular apertures of different diameters) and 72 trials in Experiment 2 (one small passable and two larger non-passable apertures). We scored first approach, first penetration attempt, and full passage; data were analyzed with generalized linear models. In Experiment 1, choices were random, unaffected by aperture size or position. In Experiment 2, first approaches were random, but first penetration attempts—and ensuing passages—were directed almost exclusively to the single passable aperture. These results indicate near-field formation of pass/not-pass judgments, likely via tactile and hydrodynamic sensing. The behavioral dissociation between exploratory (epistemic) and goal-directed (pragmatic) actions supports a modular model of self-representation, where distinct sensorimotor loops underlie information gathering and goal execution. Thus, reedfish demonstrate body-size awareness and contribute to comparative evidence that modular self-representation and embodied anticipation may extend deep into vertebrate evolution.

## 1. Introduction

By *self-representation* we refer to the ability of an organism to construct and use an internal, multimodal model of itself—integrating interoceptive, exteroceptive, and proprioceptive information within sensorimotor loops—to distinguish its own body and actions from those of other agents and from the surrounding environment [1,2,3,4,5,6]. Within this broad framework, *body size awareness* denotes a specific and operationally testable form of bodily self-representation, in which an individual takes its own physical parameters (size, shape, and mass) into account when planning or executing goal-directed interactions with the environment [1,3]. Evidence for this mechanism in taxa phylogenetically distant from humans argues against a strictly binary view of self-awareness [3,4] and even against a single gradualist ladder culminating in humans [4,5]. In line with the *modular cognition* approach, we regard self-representation as an array of functionally specialized, partially independent cognitive components that evolved to meet different ecological demands—such as monitoring one’s own body boundaries, mapping the relation between self and environment, and distinguishing self-generated from external actions [6,7]. This modular framework shifts the focus from mirror-based assays to embodied, ecologically valid challenges (e.g., matching the body to obstacles), where adaptivity-based hypotheses and cross-species comparisons are appropriate.

The principal behavioral approach to assess “body size awareness” is the body-as-obstacle paradigm: an animal is confronted with a barrier containing openings (or other narrow passages) of varying size and shape, and researchers evaluate whether it can prospectively—before physical contact—select the only passable option by matching its own body dimensions to the geometry of the obstacle [6,8,9]. Canonical implementations present choices among several apertures where only one is traversable (sometimes of smaller area), while configurations are alternated to minimize learning effects [2,10]; comparable procedures have been used with birds required to pass through holes of different height/width to reach a reward [10]. Using variants of body-as-obstacle, body size awareness has been demonstrated in children (18–26 months) [9], chimpanzees (*Pan troglodytes*) and gorillas (*Gorilla gorilla gorilla*) [11] domestic dogs (*Canis familiaris*) [6,12] and cats (*Felis catus*) [13] (sequentially decreasing apertures), budgerigars (*Melopsittacus undulatus*) [14] and bumblebees (*Bombus*) [15] (during flight through apertures), hooded crows (*Corvus cornix*) [16], domestic ferrets (*Mustela putorius furo*) [2], Wistar laboratory rats (*Rattus norvegicus domestica*) [10] (choosing the sole passable hole prior to contacting the partition), radiated ratsnakes (*Elaphe radiata*) [17] and hermit crabs (*Coenobita compressus*) [18].

In this study, we aim to test the presence of body size awareness in reedfish (*Erpetoichthys calabaricus*). This species is a suitable model due to its lifestyle and sensorimotor profile: reedfish are benthic, predominantly crepuscular/nocturnal predators that exhibit benthic feeding in water and make regular terrestrial excursions under laboratory conditions; activity increases under low light, and high-speed recordings document infrared-lit benthic and terrestrial prey capture with a “lifted-trunk” mechanism, alongside water-to-land transitional locomotion [19]. Together with documented withdrawal/fast-start responses to head/tail stimulation, this indicates a sensorimotor specialization for near-field, head-first interactions with substrates and obstacles—highly relevant to “body-as-obstacle” tasks [20]. Available evidence on polypterid sensory systems further supports this view: the closely related *Polypterus senegalus* shows very low spatial visual acuity in optokinetic tests (0.05–0.075 cycles/deg), whereas the cranial lateral-line system is well developed (canals and neuromasts), consistent with short-range mechanoreception; moreover, nocturnal amphibious ray-finned fishes display convergent expansions of the olfactory system at both receptor and processing levels [21,22,23]. Finally, behavioral work on reedfish and polypterids documents flexible transitions between aquatic and terrestrial locomotion and terrestrial prey capture, which—together with a benthic lifestyle—strengthens the expectation that tactile/mechanosensory probing underpins assessment of passability in narrow spaces; to our knowledge, however, no study has directly tested aperture negotiation and body size awareness in reedfish [19,24]. Given their predominantly tactile and mechanosensory mode of spatial perception, reedfish provide an excellent model for studying how animals with limited visual input assess the relation between their own body dimensions and environmental constraints.

## 2. Materials and Methods

### 2.1. Animals

We tested eight reedfish *Erpetoichthys calabaricus* (4 males, 4 females; total length, TL, 32–34 cm) obtained from Aqua-Logo (Moscow, Russia). Sex was determined by anal-fin morphology (males showing a broader, more fleshy anal fin) [25]. All fish were group-housed in a single 500 L glass tank with a tight-fitting lid; freshwater was maintained at 24–28 °C and pH 6.5–7.5 with 10–15% weekly water changes. The bottom substrate consisted of fine rounded gravel, with floor-level shelters (PVC pipes/caves, driftwood). Continuous access to the air–water surface was ensured. Filtration was moderate with gentle flow (external canister filters). Lighting was dim (≈100 lx at the water surface) under a 12L:12D photoperiod. Fish were fed Gammarus and bloodworms three times per week.

All animal procedures complied with the ARRIVE guidelines and the European Union Directive 2010/63/EU [26] on the protection of animals used for scientific purposes. The study protocol was reviewed and approved by the Ethics Committee of the Moscow Institute of Psychoanalysis.

### 2.2. Experimental Setup

The experimental apparatus consisted of a glass tank (800 × 400 × 400 mm; water depth 300 mm) with a tight-fitting lid; the substrate, water parameters, and filtration matched the pre-experimental housing conditions. The setup was placed in an isolated room to minimize extraneous light and noise.

An opaque black full-height partition (4 mm thick) made of non-transparent glass was positioned at mid-length, dividing the tank into two equal square compartments (“Start” and “Finish”). The partition contained three arched apertures: circular openings of 60 mm diameter truncated at the bottom by a straight chord (aperture height 50 mm); the flat edge faced downward and was positioned 10 mm above the tank bottom. Each aperture could be individually occluded with an opaque acrylic insert (square plates, 100 × 100 mm) that slid into guide slots located on the “Finish” side of the partition, allowing rapid configuration changes between trials (see Figure 1).

In the “Finish” compartment, lighting matched the baseline housing conditions; a tubular shelter (300 mm in length, 120 mm inner diameter) was centered across the width and oriented with its opening toward the partition with the apertures.

The “Start” compartment was kept under constant dim illumination of approximately 100 lx at the water surface, with no shelters. Video was recorded using a Panasonic HC-V260 camcorder (Panasonic Holdings Corp., Petaling Jaya, Selangor, Malaysia) mounted on a tripod behind the “Finish” compartment and aimed to capture the apertures in the partition.

All tank dimensions, aperture geometries, and lighting conditions were precisely measured and recorded to allow full reproducibility of the setup.

### 2.3. General Procedure

Before testing, all three apertures in the partition were fully occluded with inserts. The focal fish was placed in the “Finish” compartment and kept there for 5 days; feeding followed the pre-experimental schedule. This period served to habituate the animal to the “Finish” compartment.

Experimental trials were conducted after the 5-day habituation, during the dark phase of the light cycle. At the start of each trial, a predefined aperture configuration was set in the partition (see “Experiment 1” and “Experiment 2” for details). The shelter tube with the fish inside was then removed from the “Finish” compartment and transferred to the “Start” compartment. The fish was gently released from the shelter, after which the shelter was returned to the “Finish” compartment and oriented with its opening facing the partition bearing the apertures.

During a trial, the fish was motivated to leave the more brightly illuminated “Start” compartment, which contained no shelters, and to pass through one of the apertures into the darker “Finish” compartment containing the shelter tube (see Figure 2). A trial was considered complete if the animal exited the “Start” compartment within 15 min. The next trial was initiated only after the fish, having entered the “Finish” compartment, re-entered the shelter tube and remained inside for at least 10 min.

Each fish completed 117 trials in total (9 training, 36 in Experiment 1, and 72 in Experiment 2), with 10–12 trials conducted per day. Initially, additional rest periods between daily sessions were planned to prevent fatigue or habituation. However, behavioral observations revealed no signs of reduced activity, avoidance, or stress throughout the testing phase. The fish consistently displayed exploratory behavior and stable response latencies. Therefore, no extra rest days were required, and all experimental sessions proceeded as scheduled.

All trials were performed sequentially for each individual; upon completion of the full set, the fish was transferred to a post-experimental holding tank, after which the next subject was introduced into the apparatus and the cycle repeated.

### 2.4. Odor Control

To preclude olfactory “track” cues across trials, all handling of panels/inserts was performed with powder-free nitrile gloves. Prior to first use, cast-acrylic panels and holders were degreased with 70% ethanol and triple-rinsed with dechlorinated water. Between consecutive trials, panels were removed, triple-rinsed with water from a separate reservoir (temperature and pH matched to the tank), blotted with resin-free lab wipes, and reinstalled in the new configuration; metal tools (tongs/keys) were likewise rinsed each time. To minimize residue buildup, we used two identical panel sets in alternation: while one set was in use, the other was air-dried for ≥30 min in a clean cabinet. Between subjects, panels were additionally flushed for 10 min under running dechlorinated water, followed by triple rinse and complete air-drying. No detergents, scented materials, or disinfectants were used near the apparatus; filtration and gentle flow were kept constant, and aperture configurations were quasi-randomized, preventing the fish from exploiting potential chemical cues.

### 2.5. Training

The training series familiarized subjects with the procedure and comprised nine trials per individual. In each trial, only one of the three apertures was fully open, and trials were arranged in a quasi-random order under two constraints: (i) in each successive trial the open hole occupied a new position relative to the previous one (left/center/right), and (ii) each position was presented exactly three times. Training was deemed complete when the fish entered the “Finish” compartment within three minutes; by the last two trials, all subjects completed the transition in under 30 s.

The number of training trials (n = 9) was defined a priori to achieve minimal yet sufficient counterbalancing and to limit animal burden: each of the three aperture positions (left/center/right) was presented exactly three times (3 × 3), providing balanced control of positional effects without overtraining. We adopted the same training dose as in our methodologically comparable paradigms in mammals, where exit latencies stabilized within this range while maintaining a moderate workload for the subjects.

### 2.6. Experiment 1

#### 2.6.1. Experimental Procedure

The aim of Experiment 1 was to determine whether reedfish preferentially choose larger apertures to access the “Finish” compartment. In pilot testing, the largest individual was able to pass through a circular opening of 15 mm in diameter. Accordingly, the smallest “passable” opening in the main experiment was set conservatively at 17 mm to avoid discomfort or injury.

Each trial presented three circular, traversable apertures of distinct diameters—17 mm (small), 22 mm (medium), and 27 mm (large)—mounted in the central partition (see Figure 3). The horizontal position of each aperture (left, center, right) varied according to a quasi-random schedule under two constraints. First, across the entire series each aperture size appeared equally often in each of the three positions, i.e., 12 presentations per position (36 trials total). Second, from one trial to the next, an aperture of a given size was not allowed to remain in the same position, thereby reducing simple positional strategies.

The trial count per series was defined a priori as 36 to yield a fully balanced 3 (size) × 3 (position) design with four repetitions per cell for each subject. This length allows us to (i) equalize marginal frequencies of sizes and positions, (ii) implement first-order carryover control (a given size does not remain in the same position across consecutive trials), and (iii) achieve sufficient group-level precision (~32 observations per cell across 8 fish) without imposing excessive burden on the animals. The same trial dose has been used in our methodologically comparable paradigms in mammals, facilitating cross-species comparability.

The complete trial sequences, including the order and combinations of apertures used in each experimental series, are provided in the Appendix A.

#### 2.6.2. Recorded Variables and Statistical Analysis

We recorded three quantitative dependent variables per trial:First approach to a given hole: the fish’s snout came within ≤15 mm of the partition at a specific aperture without making physical contact with the rim.First attempt to enter a given hole: any instance in which part of the snout penetrated the aperture.Full passage through a given hole: the fish traversed an aperture with the entire body, leaving the “Start” compartment.

All analyses were conducted in R version 4.5.0. In the overwhelming majority of cases (279 out of 288), the fish ultimately passed through the same hole it had first approached. Accordingly, in subsequent analyses the chosen hole was operationalized as the hole of full passage.

Statistical analyses were performed using generalized linear models. We initially fitted a binomial generalized linear mixed model (GLMM) with a logit link, in which the probability of hole choice (CHOSEN) was modeled as a function of hole size (small, medium, large), hole position (left, center, right), and trial number (TRIAL), with subject identity (FISH) included as a random factor. However, this model resulted in singular fit (variance of the random effect estimated as zero). Therefore, we discarded the random factor and used a generalized linear model (GLM) with fixed predictors only.

The full model was compared against the null (intercept-only) model using likelihood ratio tests and AIC. Model diagnostics were conducted with the DHARMa package, including tests for dispersion, residual uniformity, and temporal autocorrelation. Multicollinearity was evaluated using variance inflation factors (VIF). Explained variance was quantified with McFadden’s pseudo-R^2^ and Tjur’s R^2^. Post hoc pairwise comparisons for the factors “hole size” and “hole position” were performed with Tukey’s correction.

### 2.7. Experiment 2

#### 2.7.1. Experimental Procedure

Experiment 2 assessed whether reedfish can select the single traversable aperture among three when the two alternatives are larger in area. Pilot observations showed that the largest individual could pass through a 15 mm circular opening; therefore, the minimum passable diameter in the main test was conservatively set to 17 mm to avoid discomfort.

The series comprised 72 trials of two types distinguished by hole shapes: 24 test trials and 48 background trials. In test trials, the partition contained one small passable aperture and two large non-traversable ones. Small passable apertures were: (b1) a circle Ø 17 mm and (b2) a 17 × 17 mm square (Figure 4). Large non-traversable apertures had an area equal to 154% of the Ø 17 mm circle and came in two variants: (a3) a horizontal rectangle 50 × 7 mm and (a4) a vertical rectangle 7 × 50 mm. Test trials were counterbalanced according to two rules: (i) from one test trial to the next, the position of the small passable hole (left/center/right) always changed; (ii) the shapes and their pairings alternated evenly so that each of the four combinations occurred six times: circle + horizontal rectangles; circle + vertical rectangles; square + horizontal rectangles; square + vertical rectangles. This prevented reliance on simple shape-based heuristics.

In background trials, the configuration was inverted: the partition held two panels with large passable apertures and one with a small non-traversable aperture. Small non-traversable shapes were: (a1) a circle Ø 7 mm and (a2) a 7 × 7 mm square; large passable shapes were: (b3) a horizontal rectangle 50 × 17 mm and (b4) a vertical rectangle 17 × 50 mm (Figure 4). Background trials were counterbalanced by (i) changing the position of the small non-traversable aperture from trial to trial, and (ii) alternating shape pairings evenly so that each of the four combinations occurred twelve times (circle + horizontal; circle + vertical; square + horizontal; square + vertical).

Across the series, two background trials always preceded one test trial, and the sequence started with background trials. This alternation was designed to prevent the formation of a simple rule to pass either the smallest hole or, conversely, the largest ones relative to their neighbors.

The total of 72 trials in Experiment 2 (24 test, 48 background) was defined a priori to achieve full counterbalancing of shapes and positions with a minimal yet sufficient data volume and moderate animal burden. Test trials implemented four combinations (small passable: circle/square × large non-traversable: horizontal/vertical rectangles), each presented six times (24 tests), with the position of the small passable hole changing from trial to trial and being evenly distributed across left/center/right. Background trials mirrored this structure with four combinations (small non-traversable: circle/square × large passable: horizontal/vertical rectangles), each presented twelve times (48 backgrounds), under the same positional control. The fixed sequence of two background trials followed by one test trial (series starting with background) prevented learning a simple “always choose the smallest” or “always choose the largest” rule, thereby targeting body size awareness rather than heuristic responding. This design yields six observations per combination at the individual level (and 48/96 at the group level across eight fish), providing adequate precision for GLM estimation and Tukey post hoc contrasts while maintaining comparability to our previous protocols.

The complete trial sequences, including the order and combinations of apertures used in each experimental series, are provided in the Appendix A.

#### 2.7.2. Recorded Variables and Statistical Analysis

We quantified three dependent measures per trial:First approach to a specific aperture (snout within ≤15 mm of the partition without contacting the rim);First penetration attempt at a specific aperture (partial insertion of the snout into the hole);Full passage through one of the apertures (the fish entirely leaves the “Start” compartment).

Operational details follow Section 2.6.

Data from the second experiment were analyzed using generalized linear models (GLMs) with a binomial error structure and logit link function. Two sets of analyses were conducted: (1) for the first penetration attempts made by the fish, and (2) for the first approaches towards an aperture within each trial. In both cases, the dependent variable was a binary outcome (0/1), indicating whether a given aperture was chosen in the respective context (attempt or approach).

Fixed predictors included aperture penetrability (b1, b2 = penetrable; a3, a4 = non-penetrable), small-hole shape (round b1 vs. square b2), large-hole shape (horizontal a3 vs. vertical a4), aperture position (Left, Center, Right), and trial number (continuous covariate). Initial models specified fish identity as a random effect (GLMM), but these models were singular (random-effect variance ≈ 0). Therefore, final analyses were conducted using GLMs with fixed effects only.

For each model, we evaluated: likelihood ratio tests against the null model, Akaike’s Information Criterion (AIC), pseudo-R^2^ measures (McFadden’s and Tjur’s R^2^), dispersion, residual uniformity, and temporal autocorrelation using DHARMa simulations, multicollinearity diagnostics (variance inflation factors, VIF).

Post hoc analyses were conducted using the emmeans package, including contrasts comparing penetrable vs. non-penetrable apertures, small-hole shape (round vs. square), large-hole shape (horizontal vs. vertical), and pairwise Tukey-adjusted comparisons across the three positions.

### 2.8. Use of AI-Assisted Tools

During manuscript preparation, we used ChatGPT 5 (OpenAI) solely for translating draft Russian text into English and for minor language editing (style/wording adjustments without altering the scientific content). All sections, data, conclusions, and interpretations were written and validated by the author; any machine-generated suggestions were reviewed and edited by a human. No AI tools were used for data generation/processing, statistical analyses, figure preparation, literature selection, or research decision-making.

## 3. Results

### 3.1. Experiment 1

The binomial GLM revealed that neither hole size, hole position, nor trial number had any significant effect on fish choices. All predictor coefficients were close to zero (all *p* > 0.79), whereas the intercept corresponded to a choice probability of ~0.33, i.e., chance level across three alternatives. Comparison of the full and null models confirmed the absence of predictor effects (χ^2^(4) = 0.09, *p* = 0.999; ΔAIC = +7.9).

Pseudo-R^2^ values indicated no explained variance (McFadden’s R^2^ = 0.0001; Tjur’s R^2^ = 0.001). Post hoc Tukey comparisons showed no differences among hole sizes (L–M, L–S, M–S, all *p* > 0.96) or among positions (left–center, left–right, center–right, all *p* > 0.96).

Model diagnostics with DHARMa indicated no problems: residuals were uniformly distributed (*p* = 0.26), dispersion was adequate (*p* = 0.95), and no temporal autocorrelation was detected (DW = 1.79, *p* = 0.074).

In summary, fish chose among the three holes at chance level, with no detectable effect of hole size or position (Figure 5).

### 3.2. Experiment 2

In this section we report analyses for the 24 test trials only. The 48 background trials served as an anti-heuristic scaffold (inverting the “small/large” contingencies to prevent simple rule learning) and were not included in the inferential models for Experiment 2. This decision was made a priori to avoid mixing data with different task contingencies and to provide an undiluted estimate of the target effect—selection of the single traversable aperture under test configurations.

#### 3.2.1. Choice of Hole for the First Approach

The analysis of first approaches revealed no effects of any predictors. The full model did not differ from the null (LR χ^2^(6) = 0.96, *p* = 0.99), and explained-variance measures were negligible (McFadden’s R^2^ = 0.0013; Tjur’s R^2^ = 0.002).

For TYPE, no significant differences were detected: the contrast comparing penetrable (b1,b2) versus non-penetrable (a3,a4) holes was non-significant (z = 0.18, *p* = 0.86) (Figure 6). Likewise, there were no differences between the two penetrable shapes (b1 vs. b2) or between the two non-penetrable shapes (a3 vs. a4) (both *p* > 0.44). POSITION had no effect on first approaches; Tukey pairwise comparisons among left, center, and right positions indicated no differences (all *p* ≥ 0.97). TRIAL also showed no effect (*p* = 0.97).

Descriptive statistics supported a random distribution of first approaches across hole types. Mean per-fish proportions (±95% CI) were: b1 = 0.312 [0.251; 0.374], b2 = 0.365 [0.282; 0.447], a3 = 0.344 [0.303; 0.384], a4 = 0.318 [0.292; 0.344], with substantial overlap among confidence intervals. Thus, first approaches were effectively random and did not depend on hole penetrability, shape, position, or trial number.

#### 3.2.2. Choice of Hole for the First Penetration Attempt

The binomial GLM revealed that hole penetrability was the decisive factor for fish choice. Fish almost exclusively selected penetrable holes (b1 or b2), whereas non-penetrable holes (a3, a4) were virtually never chosen (Figure 7). The effect of TYPE was highly significant (χ^2^(3) = 669, *p* < 0.0001), and the full model was strongly superior to the null (ΔAIC = –657).

Post hoc contrasts confirmed that penetrable holes were chosen significantly more often than non-penetrable ones (z = 9.85, *p* < 0.0001). No differences were detected between the two penetrable types (b1 vs. b2, *p* = 0.56) or between the two non-penetrable types (a3 vs. a4, *p* = 0.57).

The factor POSITION had no influence on choice (χ^2^(2) = 1.04, *p* = 0.59), and pairwise Tukey comparisons indicated no differences between left, center, and right positions (all *p* > 0.58). The effect of trial number (TRIAL) was also non-significant (*p* = 0.92).

Model diagnostics with DHARMa indicated no problems: dispersion was adequate (*p* = 0.97), residuals were uniformly distributed (*p* = 0.67), and no temporal autocorrelation was detected (Durbin–Watson = 1.82, *p* = 0.20). Measures of explained variance showed that the model accounted for nearly all variability in fish behavior (McFadden’s R^2^ = 0.91; Tjur’s R^2^ = 0.95).

It should be added that whenever a penetration attempt targeted a penetrable hole (b1 or b2), it invariably resulted in full passage into the “Finish” compartment.

In summary, fish clearly distinguished between penetrable and non-penetrable holes, almost always selecting the penetrable option, while hole shape, position, and trial number had no effect on choice.

## 4. Discussion

### 4.1. Experiment 1

When all three apertures were traversable and varied in diameter and position, reedfish showed no preference for size or location: choices were at chance. In other words, the operative principle here is not “the bigger, the better,” but “sufficient to pass.” Given equally traversable options, fish did not maximize aperture width, indicating a strategy that matches bodily limits to the minimally required resource (passability) rather than seeking redundant clearance.

A comparison with taxonomically distant mammals tested in a methodologically similar paradigm highlights cross-species differences. Ferrets consistently preferred the central hole irrespective of size, interpreted as selecting the shortest path through the partition [2]. In contrast, rats tended to avoid the center and favored the left or right hole, which aligns with their tendency to minimize exposure to open, less protected spaces [10]. In our study, neither a central nor a peripheral bias emerged—reedfish distributed choices evenly.

We propose that, when several options are equally passable, reedfish effectively adopt a “first available contact” strategy: they pass through the first aperture they approach. This pattern accords with the species’ sensory profile—low visual spatial acuity and reliance on near-field cues [21,22,23]—and we elaborate on these mechanistic underpinnings in the discussion of Experiment 2.

### 4.2. Experiment 2

Experiment 2 revealed a clear phase split. First approaches were randomly distributed across apertures, whereas the first penetration attempt (followed invariably by full passage) was directed to the single traversable aperture—even though it was smaller in area than the other two. In short, exploration showed no selectivity, while the decision to penetrate was virtually error-free.

This pattern supports a two-stage process: the “pass/not pass” judgment emerges at near range, immediately prior to action [7]. It aligns with a modular account of self-representation: (i) a scouting/orientation module and (ii) a body-to-obstacle matching module, recruited in different phases of the same goal-directed sequence [27,28]. Given the polypterid sensory profile (low visual spatial acuity and high reliance on near-field mechanosensory/tactile cues [21,22,23]), the transition from distal scanning to contact with the aperture rim is precisely where the geometry of the head/body is matched to the opening.

This two-phase organization maps naturally onto epistemic versus pragmatic actions [29,30]. Our first approaches are epistemic acts: they do not aim to achieve the outcome per se but to gather information—any approach “at random” yields tactile/hydrodynamic input sufficient to assess passability. By contrast, first penetration attempts are pragmatic acts, i.e., the use of accumulated information to attain the goal (entry into the target compartment): at this stage, choice ceases to be random, and fish reliably select the traversable aperture. Taken together, the data provide direct evidence that reedfish take their own body limits into account when choosing among competing openings.

This behavioral dissociation between exploratory (epistemic) and goal-directed (pragmatic) actions can be interpreted through the lens of embodied and predictive frameworks. In predictive-processing terms, epistemic actions minimize uncertainty about affordances, while pragmatic actions minimize expected free energy during goal attainment [30]. Within an embodied cognition perspective, these two phases correspond to distinct control loops within sensorimotor coupling—an information-gathering loop and an execution loop—which jointly constitute an adaptive body-scaled interaction with the environment [31]. This interpretation aligns with the modular account of self-representation proposed in this study: epistemic and pragmatic loops can be viewed as separable yet interacting modules that jointly support body-size awareness and action control.

#### Controlling Alternative Explanations

First, the design precluded learning of simple rules. A fully balanced 3 × 3 “size × position” plan with four repetitions per cell and quasi-random reassignment of apertures across consecutive trials prevented strategies such as “always center,” “always edge,” or “always the largest.” The null effect of TRIAL and the lack of differences across positions and sizes in the model confirm the absence of trend learning and sequence heuristics.

Second, lateral biases were explicitly tested and ruled out: no left–center–right effects were observed. Finally, routine odor/trace controls (panel rotation and rinsing, stable flow) minimized potential chemical cues. Together, these controls support the interpretation that fish were genuinely indifferent to superfluous clearance when all options were traversable, rather than being driven by procedural artifacts.

### 4.3. General Discussion

Our findings fit within the expanding body of research on body-size awareness while highlighting species-specific adaptations shaped by sensory ecology and morphology. To facilitate cross-species comparison, we organize the discussion along three heuristic dimensions: (i) sensory ecology, the dominant information channels and spatial scale of sensing; (ii) action policy, the relative balance between exploratory (epistemic) and goal-directed (pragmatic) control; and (iii) evolutionary and eco-morphological context, the niche constraints and body plan that shape passability judgments.

#### 4.3.1. Sensory Ecology

In studies employing the same or closely related “body-as-obstacle” paradigms with warm-blooded vertebrates—rats [10], ferrets [2], and crows [16]—subjects typically directed their first approach, first attempt, and subsequent passage immediately to the single traversable aperture. In contrast, reedfish displayed random first approaches but almost exclusive selection of the passable hole on the first penetration attempt. This two-phase organization—non-selective scanning followed by decisive discrimination—suggests that passability is evaluated only at near range.

Such divergence is consistent with differences in sensory resolution. Visual acuity is high enough in rats (~0.9–1.0 cpd [32]), ferrets (~0.65 cpd [33]), and corvids (~8.4 cpd [34]) to allow remote visual assessment of passability. In contrast, related polypterids exhibit extremely low acuity (~0.05–0.075 cpd in *Polypterus senegalus* [21]). Consequently, reedfish likely rely on tactile, lateral-line, and proprioceptive inputs to assess aperture size only at close distance. This sensory profile explains the random exploratory approaches and the near-deterministic choice at the attempt stage.

In reedfish, visual acuity is likely insufficient for remote passability assessment, suggesting that near-field mechanosensory systems play a major role. Based on polypterid morphology and behavior, the most plausible contributors are cranial lateral-line sensors, tactile receptors around the head and mouth, and hydrodynamic feedback generated during close inspection of the apertures [20,22,35,36]. These modalities likely operate in concert, providing real-time estimates of spatial constraints through water displacement and surface contact. Although the present study was not designed to isolate these components, future experiments could systematically test their relative contributions—for instance, by temporarily attenuating lateral-line input, manipulating water flow, or modifying aperture texture—to determine the specific mechanistic basis of passability judgments.

#### 4.3.2. Action Policy

Within a cognitive framework, this behavioral pattern supports a modular view of self-representation: behavior separates into epistemic actions (initial approaches serving information gathering) and pragmatic actions (penetration attempts and passages that utilize the acquired information) [29,30]. Reedfish express these phases sequentially within the same episode, whereas in visually oriented species the epistemic phase is externalized—it occurs before approach and relies primarily on vision. Thus, differences in sensory architecture and ecological niche (benthic, crepuscular lifestyle in reedfish) can account for cross-species variation in temporal dynamics of body-size–related decisions, while preserving the shared underlying mechanism—evaluating bodily limits to select a passable route.

#### 4.3.3. Evolutionary and Eco-Morphological Context

Our results converge with data on *Elaphe radiata* [17], obtained in a similar two-compartment, three-aperture setup. Snakes developed body-limit awareness and behavioral flexibility, but only after series-level learning. Reedfish, in contrast, displayed the same functional outcome—matching body size to aperture constraints—without extended training, relying on near-field sensory evaluation. Both species reach the same cognitive goal through different temporal and sensory pathways.

Despite methodological differences, these and related paradigms all operationalize the same construct: judging “passable vs. non-passable” as a body-scaled affordance. Therefore, the construct validity of cross-species comparison remains robust.

Comparable evidence in toddlers (door-choice task [9]) shows developmental progression from body-representation errors to accurate scaling of one’s own body dimensions. In dogs [12], body-size representation guides route selection between a too-small door and a detour, with decisions made largely without trial-and-error. Reedfish achieve an analogous outcome on a different sensory footing: random exploratory approaches followed by precise, pragmatic attempts at the traversable aperture—behavioral proof of self-scaling at the moment of action.

Similar discriminative behavior appears in single-aperture paradigms with dogs [6,12], cats [13], birds [14], and insects [15], where individuals anticipate passability before entering narrow spaces. Despite their low vision (~0.05–0.075 cpd [21] vs. 7–10 cpd in dogs [37], 7.7–10 cpd in budgerigars [38], ~3–9 cpd in cats [39] and ~0.21 cpd in bumblebees [40]), reedfish achieve the same functional outcome as visually guided taxa—accurate matching of body boundaries to environmental constraints.

Finally, convergence extends beyond vertebrates. In terrestrial hermit crabs (Coenobita compressus [18]), individuals preferentially select shells whose external geometry allows escape through a gauged opening—an “extended body” version of the same self-scaling principle. Across these diverse taxa, body-size awareness emerges as a general solution to the problem of adaptive interaction between body and environment, implemented through different sensory modalities and morphologies.

From an ecological standpoint, body-size awareness in reedfish is likely an adaptive specialization linked to their benthic and crepuscular lifestyle. *Erpetoichthys calabaricus* frequently occupies dense submerged vegetation, rock crevices, and burrow systems in slow-flowing West African waters, where successful navigation depends on precise body–environment matching. The ability to evaluate passability at near range would facilitate access to refuges, prey pursuit within confined spaces, and rapid retreat from predators. This interpretation accords with the species’ elongated morphology and amphibious behavior, which together demand continuous recalibration between aquatic and aerial locomotion [19,24]. Similar ecological pressures—predator avoidance and use of narrow shelters—have been proposed as selective contexts for body-size awareness in other taxa [1,10]. Therefore, in reedfish, the observed two-phase sensorimotor strategy likely reflects an ancient, ecologically grounded mechanism of embodied spatial cognition supporting survival in structurally complex habitats.

### 4.4. Practical Implications and Potential Applications

The present findings may have relevance beyond basic comparative cognition. First, understanding body-size awareness in non-mammalian vertebrates contributes to refining welfare practices for benthic and semi-amphibious fish in laboratory and aquaculture settings. Recognizing that reedfish adjust behavior according to body–environment fit suggests that enclosure design should provide spatial affordances allowing animals to exercise such sensorimotor judgments, reducing stress and injury during exploration or maintenance. Second, these data may inform bio-inspired robotics, particularly the development of soft or flexible robots operating in cluttered or aquatic environments. The two-phase control strategy observed in reedfish—initial exploratory (epistemic) sampling followed by precise, goal-directed (pragmatic) adjustment—parallels adaptive sensorimotor loops currently modeled in embodied robotics and active perception systems [41,42,43]. Translating these behavioral principles into robotic architectures could enhance navigation, self-calibration, and obstacle negotiation in confined or dynamic settings.

### 4.5. Limitations of the Study

First, the generalizability of our findings is constrained by the sample size and composition: eight laboratory-housed individuals from a single supplier, with a balanced sex ratio (4♂/4♀) but a narrow size range (32–34 cm TL) and no systematic coverage of age/maturity. Because the random-effects model yielded a singular fit (FISH variance ≈ 0), we adopted GLMs with fixed effects, which reduces our ability to formally quantify between-subject variability.

Second, our conclusions about the sensory basis (near-range mechanosensory/tactile dominance under low visual acuity) are inferential. We did not directly manipulate the lateral line/olfaction/vision nor did we measure visual acuity in reedfish within this protocol. Despite odor/track controls (panel rinsing, constant flow, alternating panel sets), micro-hydrodynamic or chemical cues cannot be entirely excluded; future work should instrumentally monitor microflow and include “carryover” controls for residual odor cues.

Third, the task geometry was limited: fixed water depth, three apertures, and a finite set of shapes/sizes. We did not derive a psychometric passability function around the 13–16 mm threshold, nor did we vary illumination, background contrast, temperature, flow speed, or camera vantage, any of which could modulate approach strategy. The motivational context favored a single outcome (leaving a brighter compartment for a shelter); alternative reinforcers (e.g., food) and fatigue across 10–12 trials/day were not systematically assessed. Finally, we did not implement formal observer blinding in video scoring (if this is accurate; replace accordingly if blinding was used), leaving a residual risk of observer bias.

Together, these constraints do not undermine the central pattern (chance performance when all apertures are traversable; near-perfect targeting of the single traversable aperture at the attempt stage), but they delineate clear avenues for follow-up studies: larger and more diverse samples, direct sensory manipulations, richer stimulus spaces, psychophysical thresholding, and strengthened controls for potential microcues and observer effects.

## 5. Conclusions

Reedfish (*Erpetoichthys calabaricus*) exhibited behavioral evidence of body size awareness in a “body-as-obstacle” task. When all apertures were traversable (Experiment 1), choices were at chance, indicating no maximization of aperture size under equal passability. In contrast, with a single traversable option (Experiment 2), first approaches were exploratory and random, whereas the first penetration attempt—and the ensuing full passage—was directed almost exclusively to the traversable hole.

Taken together, these findings indicate that the pass/not-pass judgment is formed at near range, relying on sensorimotor cues that match head/body limits to environmental constraints. This aligns with a modular view of self-representation: reedfish behavior separates epistemic actions (information gathering) from pragmatic actions (using information to achieve the goal). Thus, we extend the comparative landscape of body size awareness to a taxonomically distant, benthic fish with low visual acuity.

Methodologically, our strictly counterbalanced three-aperture paradigm robustly isolates the “passability assessment” module from potential heuristics (position, shape, order). Future work should quantify passability thresholds (psychophysical functions) and manipulate sensory channels (e.g., lateral-line perturbations, illumination) to directly test the near-field mechanism implied by our data.

## Figures and Tables

**Figure 1 animals-15-03231-f001:**
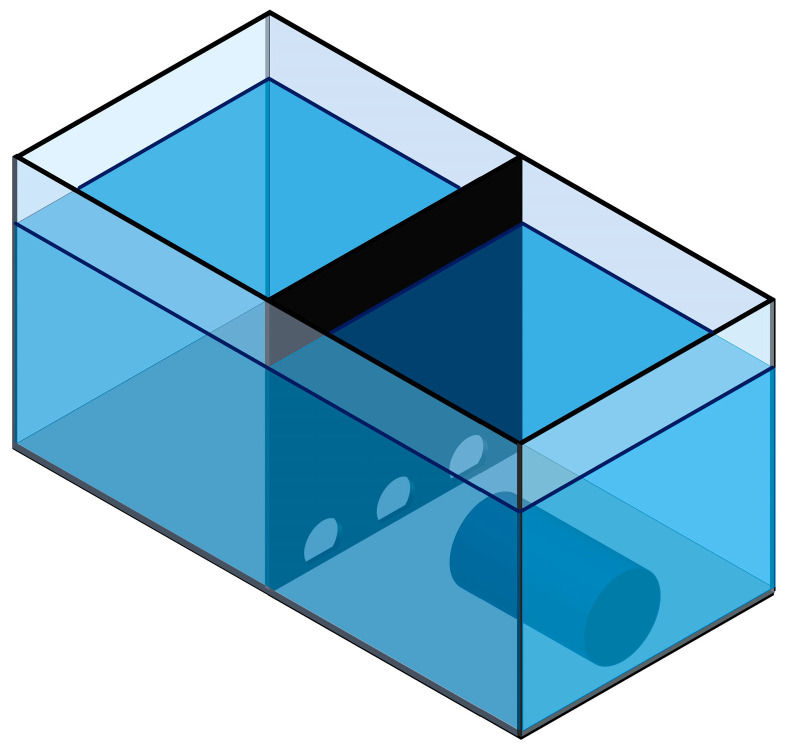
Experimental apparatus (isometric view): glass tank 800 × 400 × 400 mm with 300 mm water depth; an opaque midline partition divides the tank into “Start” and “Finish” compartments. The partition contains three arched apertures positioned 10 mm above the bottom: circular openings of 60 mm diameter truncated at the bottom by a straight chord (aperture height 50 mm), with the flat edge facing downward. A tubular shelter (300 mm in length, 120 mm inner diameter) is placed in the “Finish” compartment with its opening toward the partition. See Section 2.2 for details.

**Figure 2 animals-15-03231-f002:**
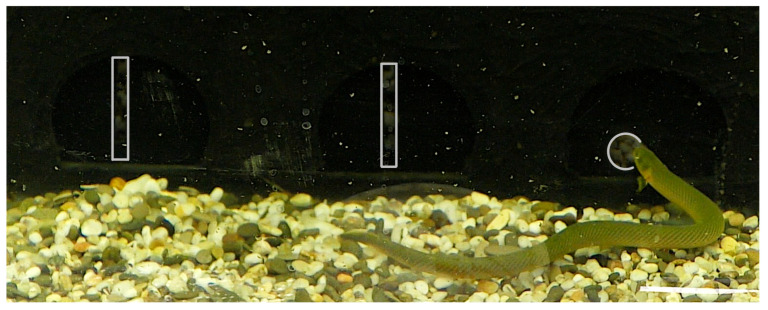
Representative video frame of the experiment: light overlays indicate the three apertures in the partition—two large non-traversable vertical openings (left and center) and one smaller traversable circular opening (right); the frame shows the fish approaching the traversable aperture (see Section 2.7.1 for details).

**Figure 3 animals-15-03231-f003:**
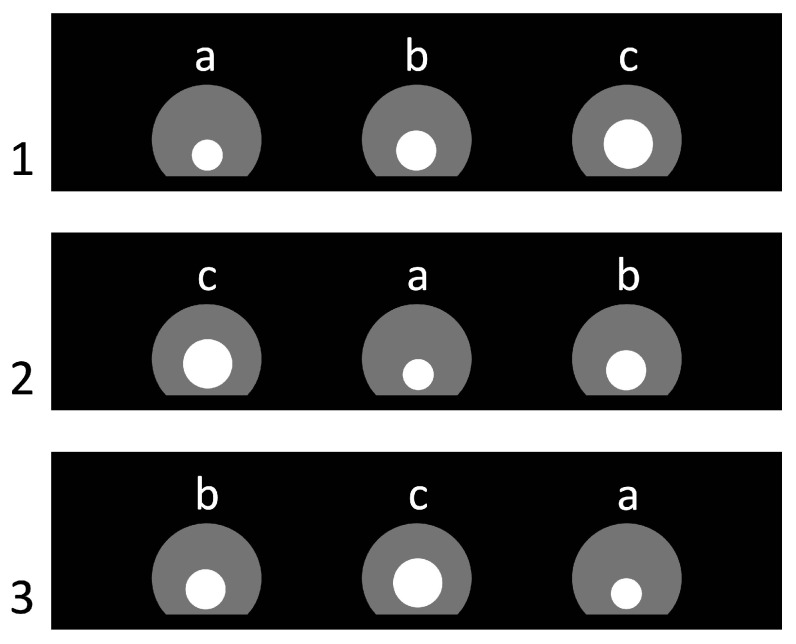
Arrangement of the inserted panels with holes in the first three trials of Experiment 1. Trial numbers are shown on the left. Letters indicate the types and sizes of the round holes: a—small (Ø 17 mm), b—medium (Ø 22 mm), and c—large (Ø 27 mm). Semicircular apertures of maximal size in the partition are depicted in gray; the white circles mark the holes actually used in each trial.

**Figure 4 animals-15-03231-f004:**
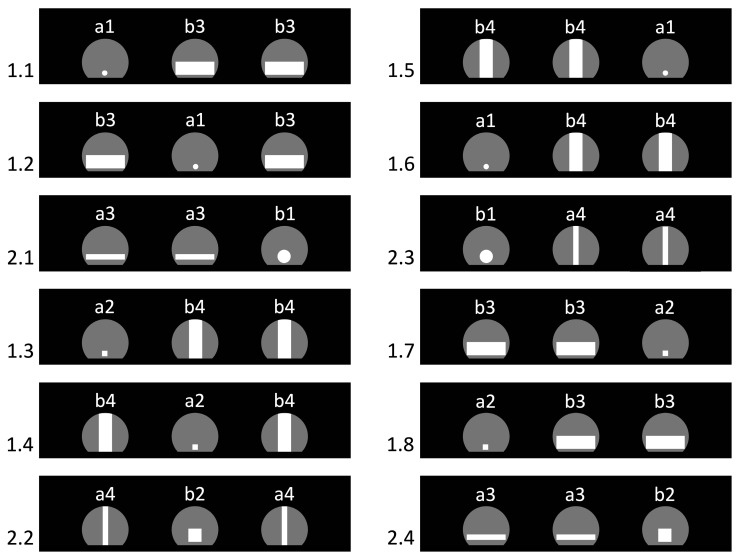
Arrangement of the hole panels in the first 12 trials of Experiment 2. The numbers on the left denote trial type and order: 1.n—background trials; 2.n—test trials. Letter–number codes on the panels indicate hole types (a—non-traversable; b—traversable; for detailed definitions see Section 2.7.1). Gray shapes depict the partition’s maximal-size arched apertures, while white shapes mark the holes actually used in each trial.

**Figure 5 animals-15-03231-f005:**
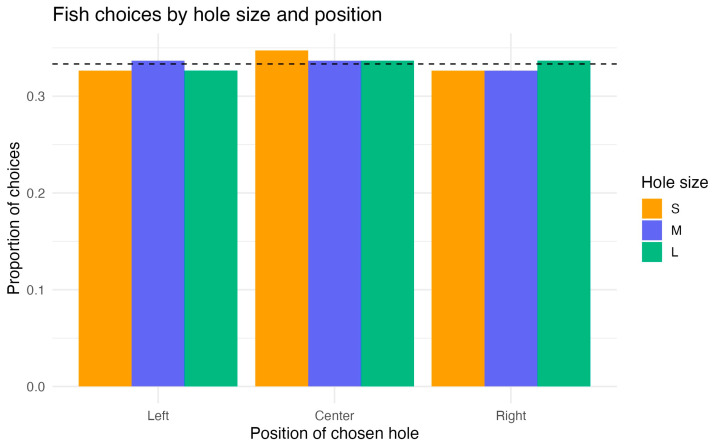
Experiment 1: proportion of choices by hole position (Left/Center/Right) and size (small/medium/large). Bars show per-fish means with 95% CIs; the dashed line marks chance level (1/3). Neither hole size nor position affected choice (see Section 3.1).

**Figure 6 animals-15-03231-f006:**
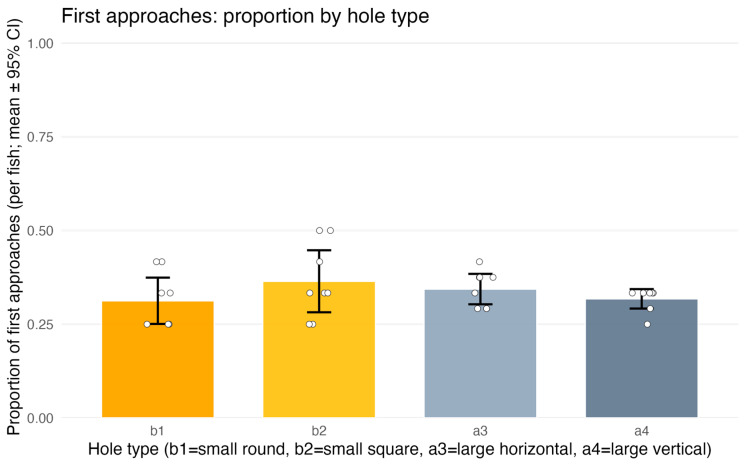
Experiment 2 (test trials only)—first approaches to apertures: proportion of approaches by hole type. b1 and b2 denote penetrable openings (circle Ø17 mm; square 17 × 17 mm); a3 and a4 are non-traversable (horizontal rectangle 50 × 7 mm; vertical rectangle 7 × 50 mm). Bars show per-fish means with 95% CIs; dots represent individual fish. First approaches were distributed at chance level and did not depend on aperture type (see Section 3.2.1).

**Figure 7 animals-15-03231-f007:**
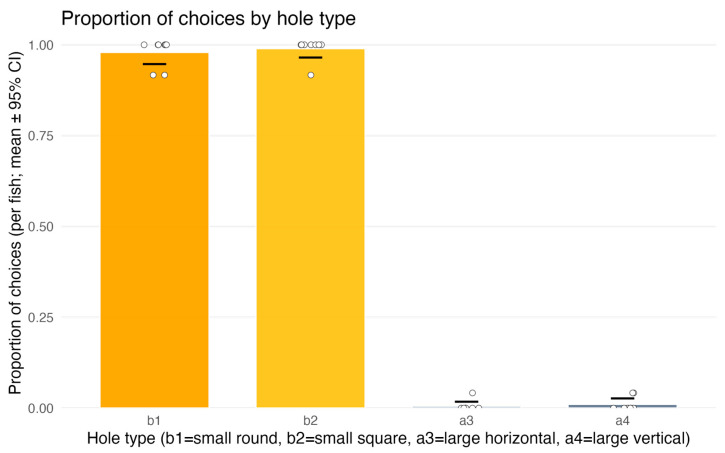
Experiment 2 (test trials only)—first penetration attempts: proportion of choices by aperture type. b1 and b2 denote penetrable holes (circle Ø17 mm; square 17 × 17 mm), a3 and a4 are non-penetrable (horizontal rectangle 50 × 7 mm; vertical rectangle 7 × 50 mm). Bars show per-fish means with 95% CIs; dots represent individual fish. Fish directed their first attempt almost exclusively to the penetrable apertures (b1, b2), whereas attempts toward non-penetrable ones (a3, a4) were near zero (see Section 3.2.2).

## Data Availability

To obtain the data, please contact the corresponding author.

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
