# Peer review of "Body Size Awareness and Modular Self-Representation in Reedfish (*Erpetoichthys calabaricus*): Near-Field Passability Judgments"

_animals, 2025, doi:10.3390/ani15223231_

Round 1

Reviewer 1 Report

Comments and Suggestions for Authors

The manuscript presents a well-designed behavioral study investigating body size awareness in Erpetoichthys calabaricus and clearly contributes to the comparative literature on self-representation mechanisms in non-mammalian species. The methodology is well organized, and the results are clearly described. The paper is overall of good quality, but some aspects could be clarified or slightly refined to improve clarity and readability. Specific comments are provided below.

  1. Line 70–89 (Introduction): I suggest to add a short bridging sentence to better connect the description of reedfish sensory ecology with the experimental rationale. This would help the reader understand why this species is particularly suited for investigating body-size awareness.
  2. (Matherial and Methods): provide a reference supporting the use of anal-fin morphology for sex identification in E. calabaricus.
  3. Section 2.3(General Procedure): Each fish completed 117 trials, which represents a substantial workload. It would be helpful to specify whether rest periods were scheduled between trial blocks or testing days, and how potential fatigue or habituation effects were minimized or monitored. Including this information would improve animal welfare informations and data reliability.
  4. (Discussion): The discussion is insightful and comparative, though it could be made more concise to improve readability and flow.

Author Response

The author greatly appreciate thorough reading of the text and thoughtful comments of the Reviewer.

Comments 1:

Line 70–89 (Introduction): I suggest to add a short bridging sentence to better connect the description of reedfish sensory ecology with the experimental rationale. This would help the reader understand why this species is particularly suited for investigating body-size awareness.

Response 1:

We thank the reviewer for this helpful suggestion. Following their advice, we added a bridging sentence to improve the logical flow between the description of the reedfish sensory ecology and the experimental rationale. The new sentence emphasizes why this species represents an appropriate model for studying body-size awareness. (Lines 119–122).

Comments 2:

(Matherial and Methods): provide a reference supporting the use of anal-fin morphology for sex identification in E. calabaricus.

Response 2:

A reference has been added. (Lines 127 and 838-839)

Comments 3:

Section 2.3(General Procedure): Each fish completed 117 trials, which represents a substantial workload. It would be helpful to specify whether rest periods were scheduled between trial blocks or testing days, and how potential fatigue or habituation effects were minimized or monitored. Including this information would improve animal welfare informations and data reliability.

Response 3:

We appreciate the reviewer’s attention to animal welfare and experimental reliability. Following their comment, we added a clarification regarding rest periods and monitoring of potential fatigue effects (Lines 197-203). As stated in the revised text, additional rest days were initially planned; however, no behavioral signs of fatigue, avoidance, or stress were observed. The fish maintained stable exploratory behavior and response times across all trials, and therefore no extra rest periods were necessary.

Comments 4:

(Discussion): The discussion is insightful and comparative, though it could be made more concise to improve readability and flow.

Response 4:

We thank the reviewer for this valuable suggestion. In the revised manuscript, the Discussion (Section 4.3) has been substantially condensed to improve readability and logical flow while retaining all relevant comparisons and citations. The revised section now presents a clear three-part structure that organizes the cross-species analysis by sensory ecology, action policy, and evolutionary–eco-morphological context, which enhances the clarity and coherence of the discussion. (Lines 521–627).

Reviewer 2 Report

Comments and Suggestions for Authors

1. The modular cognition framework requires deeper integration with contemporary theoretical debates in animal cognition, particularly regarding how the two-stage decision process (epistemic vs. pragmatic actions) relates to established cognitive architectures like embodied cognition and predictive processing frameworks.
2. The hypothesized reliance on near-field mechanosensory cues needs more detailed explanation of operational mechanisms, specifically articulating testable predictions about the relative contributions of lateral line sensing, tactile feedback, and hydrodynamic cues during aperture assessment.
3. Ecological relevance requires substantial development by explicitly connecting body size awareness to reedfish's natural benthic behaviors, including specific adaptive advantages in navigation, predator avoidance, and evolutionary context.
4. Methodological transparency should be enhanced through clearer justification of experimental parameters (aperture sizes, trial sequences) and discussion of alternative design choices to improve reproducibility.
5. Key terminology ("body size awareness," "self-representation," "modular cognition") needs precise operational definitions and consistent application throughout the manuscript to avoid conceptual ambiguity.
6. The comparative analysis would benefit from a structured framework that explicitly identifies analytical dimensions (sensory ecology, cognitive complexity, evolutionary pathways) for cross-species comparisons.
7. Practical applications in animal husbandry and bio-inspired robotics require concrete implementation examples and specific translation strategies to demonstrate real-world relevance.
8. The title and abstract should better highlight theoretical contributions to modular self-representation theories to attract broader readership in comparative cognition and evolutionary psychology.

Author Response

The author greatly appreciates the reviewer's careful reading of the text and thoughtful comments.

Comments 1:

1. The modular cognition framework requires deeper integration with contemporary theoretical debates in animal cognition, particularly regarding how the two-stage decision process (epistemic vs. pragmatic actions) relates to established cognitive architectures like embodied cognition and predictive processing frameworks.

Response 1:

We are grateful for this insightful and constructive comment. In the revised manuscript, we expanded the end of Section 4.2 to explicitly integrate the proposed modular account of self-representation with contemporary theoretical frameworks in animal cognition. Specifically, we now relate the observed two-stage behavioral organization (epistemic vs. pragmatic actions) to embodied cognition and predictive processing models. The new paragraph clarifies that epistemic actions correspond to uncertainty-reducing exploration of affordances, while pragmatic actions minimize expected free energy during goal-directed performance [Friston et al., 2015]. Within an embodied cognition perspective [Wilson, 2002], these two phases can be viewed as complementary sensorimotor control loops—information-gathering and execution—that jointly support body-size awareness and adaptive interaction with the environment. This addition provides theoretical integration without overstating the interpretive scope of the empirical data. (Lines 499–508).

Comments 2:

The hypothesized reliance on near-field mechanosensory cues needs more detailed explanation of operational mechanisms, specifically articulating testable predictions about the relative contributions of lateral line sensing, tactile feedback, and hydrodynamic cues during aperture assessment.

Response 2:

We appreciate this important and constructive comment. In the revised version, we expanded the discussion within Section «4.3.1. Sensory ecology» to clarify the hypothesized mechanosensory basis of aperture assessment in reedfish. We now specify that, given the low visual acuity of polypterids, near-field sensing likely involves the cranial lateral-line system, tactile receptors, and hydrodynamic feedback acting in concert to provide real-time spatial information. The revised text also outlines testable predictions for future research, such as experimental manipulation of lateral-line input, water flow, or aperture surface texture to evaluate the relative contributions of these sensory modalities. These clarifications retain the hypothesis-based character of the claim while improving its empirical transparency. (Lines 564–574).

Comments 3:

Ecological relevance requires substantial development by explicitly connecting body size awareness to reedfish's natural benthic behaviors, including specific adaptive advantages in navigation, predator avoidance, and evolutionary context.

Response 3:

We thank the reviewer for this insightful comment. In response, we have expanded the Evolutionary and eco-morphological context subsection (Section «4.3.3. Evolutionary and eco-morphological context») to clarify the ecological relevance of body-size awareness in Erpetoichthys calabaricus. The revised text now explicitly links this ability to the species’ natural benthic and crepuscular lifestyle, emphasizing adaptive advantages such as efficient navigation through dense vegetation, use of narrow shelters, prey pursuit, and predator avoidance. References to locomotor and ecological studies of reedfish [Pace & Gibb, 2011; Van Wassenbergh et al., 2017] and to comparative findings in other taxa [Dale & Plotnik, 2017; Khvatov et al., 2024] were added to support this interpretation. These changes strengthen the ecological and evolutionary grounding of the discussion. (Lines 615–627).

Comments 4:

Methodological transparency should be enhanced through clearer justification of experimental parameters (aperture sizes, trial sequences) and discussion of alternative design choices to improve reproducibility.

Response 4:

We appreciate this valuable comment and have revised the Materials and Methods section accordingly. In Section 2.2 (Experimental Setup), we added detailed information on the construction of the partition (4 mm opaque black glass), the shape and mounting of acrylic inserts (100 × 100 mm plates sliding into guide slots on the “Finish” side), and the rationale behind aperture dimensions and trial numbers. These design parameters were chosen based on pilot observations to ensure both behavioral reliability and animal welfare. (Lines 145-105 and 164-165).

Additionally, we now specify that full trial sequences for both experiments are presented in the Supplementary Materials to facilitate reproducibility. Together, these additions enhance methodological transparency and experimental justification. (Lines 264-265 and 342-343).

Comments 5:

Key terminology ("body size awareness," "self-representation," "modular cognition") needs precise operational definitions and consistent application throughout the manuscript to avoid conceptual ambiguity.

Response 5:

We thank the reviewer for this important remark. In response, we have revised the opening paragraph of the Introduction to include clear and operational definitions of the key terms used in the manuscript. Specifically, self-representation is now defined as a multimodal model of one’s own body integrating integrating interoceptive, exteroceptive, and pro- prioceptive information within sensorimotor loops. Body size awareness is described as a specific form of bodily self-representation in which an individual takes its own physical parameters into account when interacting with the environment. Furthermore, the concept of modular cognition is defined following Lenkei et al. (2020) as an array of functionally specialized and evolutionarily ancient components of self-representation. These clarifications improve conceptual precision and ensure consistent terminology throughout the paper. (Lines 62-69 and 74-78).

Comments 6:

The comparative analysis would benefit from a structured framework that explicitly identifies analytical dimensions (sensory ecology, cognitive complexity, evolutionary pathways) for cross-species comparisons.

Response 6:

We appreciate this insightful recommendation. The Discussion has been restructured accordingly. We introduced a concise heuristic framework that organizes the comparative analysis along three analytical dimensions: (i) sensory ecology — dominant sensory modalities and spatial scale of information acquisition; (ii) action policy — the relative contribution of exploratory/epistemic and goal-directed/pragmatic control phases; and (iii) evolutionary and eco-morphological context — body plan and niche constraints shaping passability judgments. This addition provides a clearer comparative logic without overstating theoretical claims, maintaining the empirical focus of the study. (Lines 521–627).

Comments 7:

Practical applications in animal husbandry and bio-inspired robotics require concrete implementation examples and specific translation strategies to demonstrate real-world relevance.

Response 7:

We appreciate this constructive suggestion. In response, we added a new subsection titled “Practical Implications and Potential Applications” (Section 4.4), which outlines two possible directions of applied relevance. First, we discuss how understanding body-size awareness can improve welfare and enclosure design for benthic fish in laboratory and aquaculture environments. Second, we highlight the translational potential of the observed two-phase (epistemic–pragmatic) control strategy for bio-inspired and embodied robotics, referencing recent works in biomimetic systems [Cowan & Fortune, 2007; Pfeifer & Bongard, 2006; Prescott et al., 2019]. These additions clarify how the present study extends beyond theoretical contribution to practical domains. (Lines 705–718).

Comments 8:

The title and abstract should better highlight theoretical contributions to modular self-representation theories to attract broader readership in comparative cognition and evolutionary psychology.

Response 8:

We thank the reviewer for this valuable suggestion. The title has been changed to highlight the study's theoretical contribution to modular models of self-representation: "Body Size Awareness and Modular Self-Representation in Reedfish (Erpetoichthys calabaricus): Near-Field Passability Judgments" (lines 2–4).

The abstract has been rewritten to clearly integrate the results into the context of modular cognition, emphasizing the dissociation between epistemic (exploratory) and pragmatic (goal-directed) actions as evidence for the existence of distinct sensorimotor modules in self-representation. These changes strengthen the conceptual framework and expand the study's relevance to readers engaged in comparative cognitive theory and evolutionary psychology. (Lines 24–41)
